# Community perspectives on cardiovascular disease control in rural Ghana: A qualitative study

Bhavana Patil[1☯], Isla Hutchinson Maddox[1☯], Raymond Aborigo[2], Allison P. Squires[3], Denis Awuni[2], Carol R. Horowitz[1], Abraham R. Oduro[2], James F. Phillips[4], Khadija R. Jones[1], David J. Heller[1]*

1 Arnhold Institute for Global Health, Icahn School of Medicine at Mount Sinai, New York, NY, United States of America, 2 Navrongo Health Research Centre, Navrongo, Upper East Region, Ghana, 3 Rory Meyers College of Nursing, New York University, New York, NY, United States of America, 4 Mailman School of Public Health, Columbia University, New York, NY, United States of America

☯ These authors contributed equally to this work.
* david.heller@mssm.edu

**Data Availability Statement:** Dr. Peter Wontuo, Data Manager at the Navrongo Health Research Centre, peter.wontuo@navrongo-hrc.org, is the point of contact for this data request.

## Abstract

### Background

Cardiovascular disease (CVD) prevalence is high in Ghana—but awareness, prevention, and treatment is sparse, particularly in rural regions. The nurse-led Community-based Health Planning and Services program offers general preventive and primary care in these areas, but overlooks CVD and its risk factors.

### Methods

We conducted in-depth interviews with 30 community members (CM) in rural Navrongo, Ghana to understand their knowledge and beliefs regarding the causes and treatment of CVD and the potential role of community nurses in rendering CVD care. We transcribed audio records, coded these data for content, and qualitatively analyzed these codes for key themes.

### Results

CMs described CVD as an acute, aggressive disease rather than a chronic asymptomatic condition, believing that CVD patients often die suddenly. Yet CMs identified causal risk factors for CVD: not only tobacco smoking and poor diet, but also emotional burdens and stressors, which cause and exacerbate CVD symptoms. Many CMs expressed interest in counseling on these risk factors, particularly diet. However, they felt that nurses could provide comprehensive CVD care only if key barriers (such as medication access and training) are addressed. In the interim, many saw nurses' main CVD care role as referring to the hospital.

### Conclusions

CMs would like CVD behavioral education from community nurses at local clinics, but feel the local health system is now too fragile to offer other CVD interventions. CMs believe that

**Funding:** This work was supported by the Fogarty International Center of the National Institutes of Health, USA (R21 TW010452-01 to JFP, ARO, and DJH); the Arnhold Institute for Global Health at the Icahn School of Medicine at Mount Sinai, USA (BP, IHM, KJ, DJH); the Navrongo Health Research Centre, Ghana (RA, DA, ARO); the Mailman School of Public Health, Columbia University (JFP); the Rory Myers School of Nursing at NYU (APS); Teva Pharmaceutical Industries (RA, KJ, DJH); and Resolve to Save Lives (RA, KJ, DJH).

**Competing interests:** I have read the journal's policy and the authors of this manuscript have the following competing interests: DJH, RA, and KJ report research funding from Teva Pharmaceutical Industries. DJH and KJ report consulting funding from the Hess Corporation. Neither of these funders had no role in study design, data collection and analysis, decision to publish, or preparation of the manuscript.

a more comprehensive CVD care model would require accessible medication, along with training for nurses to screen for hypertension and other cardiovascular risk factors–in addition to counseling on CVD prevention. Such counseling should build upon existing community beliefs and concerns regarding CVD–including its behavioral and mental health causes–in addition to usual measures to prevent CVD mortality such as diet changes and physical exercise.

## Introduction

In low- and middle-income countries (LMICs), there has been a seismic shift in healthcare delivery as the burden of non-communicable diseases (NCD) begins to surpass that of infectious diseases [1]. The weight of this burden is especially apparent when surveying the incidence of cardiovascular disease (CVD) in LMICs, where more than three-fourths of global CVD mortality occurs [2], and disease burden continues to rise [3, 4]. Ghana is no exception: the rate of CVD mortality is substantial in urban and rural areas [5, 6], and risk factors for CVD are increasingly common yet poorly controlled. Hypertension, the leading cause of global CVD, affects 25% to 30% of Ghanaians over 15 [7, 8], but rates of awareness, treatment, and control are under 10% [7]. Persons with non-communicable diseases such as hypertension in LMICs are often diagnosed only after showing symptoms [9], which may be too late for optimal prevention and treatment.

A lower-middle income country, Ghana has made numerous innovations to improve access to preventive care and decrease premature morbidity and mortality. In addition to a National Health Insurance Scheme (NHIS) covering more than 10 million citizens [10], Ghana has constructed a universal primary care access program for essential rural health care, the Community-Based Health Planning and Services (CHPS) initiative. CHPS was developed and tested in the 1990s by the state-funded Navrongo Health Research Centre in the remote, low-income Upper East Region of the country, as a direct response to high rates of mortality in children under 5, pregnant women, and other vulnerable groups [11, 12], and then scaled up across Ghana [13]. The program assigns nurses and volunteers to remote areas unserved by existing health clinics, where they provide front-line care both at outpost health centers called compounds, and via door-to-door visits. Yet at present, CHPS nurses are not empowered to treat CVD risk factors such as hypertension at the compound level. They instead must refer patients with elevated blood pressure to higher-level care–often at distant health centers patients cannot access [14].

Evidence from diverse global settings suggests that CHPS nurses may be able to provide such care, and offers guidelines for how to do so. In 2016, the World Health Organization published the HEARTS care model for cardiovascular disease prevention, building on prior research demonstrating that non-physician providers–including community volunteers, nurses, and pharmacists–can screen and treat for CVD risk factors such as hypertension [15]. Previous research has demonstrated that CHPS nurses can diagnose hypertension and refer patients to treatment consistently and accurately [16]–however, whether and how CHPS nurses might provide other hypertension care–through counseling, medication, or other interventions—at front-line rural compounds remains unclear [14, 17]. We, therefore, conducted a qualitative study of local perspectives on CHPS' current, potential, and ideal role in the prevention and treatment of CVD in Navrongo, designed to inform the design and community acceptance of such an intervention. We selected this site both due to its history in creating and expanding the CHPS care model, and its documented high and rising rate of CVD mortality [6].

## Methods

### Design

A qualitative descriptive design structured this study. Qualitative descriptive studies seek to gather rich, thick descriptions about a phenomenon of interest [18]. They are also useful for improving contextual understanding of environments where new interventions may be developed, tested, or adapted. Qualitative description deliberately and necessarily eschews prior presumptions, interpretations, or frameworks to interpret data–relying instead only on interview data itself, interpreted through everyday language [18] and the respondent's own words [19]. We selected this approach to avoid making assumptions about how respondents would describe their experiences and opinions regarding CVD in Navrongo given the absence of prior such research in this community. The goal of the study was to understand the participants' 1) personal experience with CVD; 2) knowledge and attitudes regarding this disease, including stigma; 3) current engagement with the CHPS care program regarding CVD; and 4) beliefs regarding how CHPS should best adapt to treat and prevent CVD, among other subjects. The study took place in the latter half of 2017.

### IRB approval

Our protocol for study recruitment was approved by the institutional review boards (IRBs) of the Icahn School of Medicine at Mount Sinai (Study #16–01601); Columbia University; and the Navrongo Health Research Centre (Study #250).

### Sample and recruitment

The targeted sample was adult community members living in villages in and around the Navrongo, Ghana area. We used convenience sampling to identify participants, aiming for a geographically diverse sample. We planned to interview approximately 30 participants in order to achieve thematic saturation, based on previous empiric research [20–22] demonstrating that roughly 12 [20] to 31 [21] interviews are sufficient to achieve thematic saturation, which we defined as the point at which new data are only repetitive of what was expressed in prior data [22].

Our aim was to understand the perceptions of the community regarding CVD (e.g., family members of persons living with cardiovascular disease) and not just those diagnosed with CVD. Therefore, we did not actively screen participants' medical history. Furthermore, prior evidence demonstrates that hypertension and other cardiovascular conditions are underdiagnosed across Ghana [8], but nonetheless very common in Navrongo [23] where they are a large and growing cause of local mortality [6], such that any attempt to limit our interviews to those already diagnosed would fail to capture most persons at risk or affected by CVD. We actively recruited an equal balance of participants identifying as male or female. We did not however record the gender identity of any given participant.

### Data collection

Before collection of any data, all interviewers were trained in person by co-authors RAA and DA, in conversations conducted chiefly in English but switching to the local languages Kasem and Nankam as necessary to clarify how to translate nuances such as how precisely to define concepts such as "heart disease," "cardiovascular disease," and what may constitute being "sick" due to such conditions. These interviewers were separately trained, as in prior studies, on appropriate approaches to probing replies using open-ended and non-leading questions. As per the interview guide [S1 Appendix], we actively avoided technical terms in this

interview–both to avoid errors in translation and to elicit participants' views in a manner most comprehensible to them. Trained interviewers obtained informed consent verbally, but documented this consent in writing–either by the participant's written signature for literate persons, or the interviewer's documentation otherwise, witnessed by the participant. No participants were minors. Our interview guide is found in [S1 Appendix]. All interviewers were staff from the Navrongo Health Research Centre (NHRC). Only these interviewers conducted each semi-structured interview, and recorded its audio. Although the guide was written in English, the interviewers conducted them in the preferred language of the interviewer–English, Kasem, or Nankam–and were themselves fluent in all three. The interview guide was not translated in writing into either of the latter two languages–rather, research assistants fluent in at least one of the local languages of the study area conducted the interview orally in that language, with the English question list as a guideline. Informed consent also always occurred in the preferred local language, using protocols validated in prior research in this community [24]. The audio recording of each such interview was then translated back into English (by the same or a different research staff member fluent in the preferred language) and then transcribed by that same member of the research team.

## Data analysis

We analyzed these data using a qualitative descriptive approach [18, 19, 25, 26]. Our goal was to identify and distill common elements and concepts across numerous interviews—in order to describe a shared set of beliefs and experiences that captures their collective outlook on the matter of adapting CHPS for CVD care [25]. This approach deliberately avoids assigning any a priori conceptual framework to the analysis [26], just as there is no framework to the interview guide [18, 19], because any initial assumed perspective might bias the way data is collected or interpreted [18, 25]. However, we nonetheless systematically applied conventional content analysis [26] to the interpretation of our data, a method validated for qualitative descriptive studies [25] on domains with little prior research [26]. In brief, two different researchers [BP and IHM] independently reviewed all written transcripts in English obtained from audio recordings of the interviews. Each wrote down their initial impressions and began to generate codes from the data inductively, i.e., based only on impressions of the data in itself. Two other researchers [KJ and DJH] also participated independently in this process, but only for select transcripts chosen at random. The research team met to compare codes and iteratively refine and generate a final codebook together.

Once this process was complete, two main reviewers [IHM and BP] separately reviewed each transcript once more, coding all written data using this agreed codebook. To ensure inter-rater reliability between these reviewers, we calculated a Kappa score of agreement between the two sets of independent codes they applied to select transcripts chosen at random, with a result of 0.68. Kappa scores by definition range from -1 to +1, with a higher score indicating greater agreement—specifically, a score of 0.68 indicates "substantial agreement." [27]. Any disagreement on any given coding element was resolved through consensus discussion between IHM and BP, with input from DJH, AS, KJ and others if needed. We performed all analyses using NVIVO software version 12 [28].

Following the completion of coding, researchers IHM, BP, DJH, and KJ independently reviewed all codes to identify cross-cutting themes that described the interviews as a whole. This thematic analysis used a social-ecological model [29] due to its capacity to describe behavioral influences ranging from immediate influences such as family to structural and societal factors such as culture, geography, or religion. We did not apply any predetermined framework to thematic analysis (like interview design or coding) for the reasons detailed above, but

our prior qualitative research in this field did inform our analyses. For instance, insofar as that work suggests that CHPS nurses believe community members lack knowledge of the causes and treatments for CVD [14, 17], we examined whether coded data seemed to support or rebut this finding.

## Results

The final sample comprised 30 community members, aged 20 through 75. Following coding using the methods detailed above, this sample proved sufficient to achieve data saturation. Key themes emerged along three topics: community members' understanding of CVD, their perceptions of what CHOs can do to treat CVD, and barriers to initiating care.

### Theme 1: CM understanding of CVD

This theme refers to CMs' understanding of CVD itself, including risk factors, prevention, and treatment. CMs attributed a wide range of symptoms to CVD—including heart and whole-body aches, difficulties breathing, feeling faint, rapid heartbeats, weight loss, confusion, and headache. In a few instances, CMs stated that heartburn is a sign of CVD. When asked about how CVD manifests in a patient, some CMs described malaria-induced fevers; others recounted the chronic fatigue of tuberculosis. Almost no CMs described CVD as asymptomatic.

*"The person [with heart disease] was persistently coughing. As a result, she lost weight, always falling sick, breathing problems and weak and eventually died."*

*"They said her heart was paining, so when her heart was paining, she could not breathe. So she too, she easily gets angry. So it happened, and she was just saying her heart is paining. And they sent her to hospital, when she got there, she died. . .you will know it was heart disease that killed her."*

*"[A] person with heart disease also [loses] weight and always feels tired, and [they] have unstable mind and [feel] depressed without knowing what exactly is wrong and [it] can lead to death."*

Almost all community members described at least one high burden risk factor for CVD, such as smoking, drinking, consuming kola nut (a caffeine-containing nut), or poor diet. Some identified chronic comorbidities such as diabetes.

*"The person asked how the disease comes about. Because he always has doubts. We discussed that the type of food eaten can cause such diseases, Smoking cigarette and alcohol intake can cause these diseases. So that was what we [the interviewee and a CHPs care member] discussed."*

In addition to these high-risk behaviors, CMs stated that noise and negative emotions are causally linked to CVD. For example, many interviewees believe that stress, anger, overthinking, and talking too much contribute to an individual's overall risk of developing CVD. Similarly, individuals believe that CVD actually causes and exacerbates these emotional stressors in a sort of feedback loop.

*"You will a feel a sharp pain from your heart like something is about to happen to you and also when you working you think a lot this makes the heart beat faster."*

*"When one thinks a lot to the extent that one is not able to sleep, [this] could lead to getting a heart disease. . .It's the too much thinking that makes people get heart diseases."*

*"Even if the person doesn't say. . .they don't hide sickness oh. . .It causes anger a lot. If the anger comes, it will show for people to know you have heart disease. It can affect you a lot. For people to know oh, where your anger has gotten to, that has resulted into that. They will know heart disease is in your body."*

In discussing prevention of CVD, interviewees emphasized amending their diet and reducing smoking and alcohol intake. Respondents suggested consuming carbohydrate-based local foods such as millet, tuo zaafi (T.Z., a hot, starchy dish), and dawadawa (locust bean) to improve their cardiovascular health. They identified oil, fatty foods, and meats as contributing to CVD. A few respondents, however, could not identify any risk factors or ways to prevent CVD. Some explained that they spent little or no time at the CHPS clinic and had not been counseled regarding CVD prevention. Additionally, screening was not mentioned in discussions about how CVD can be prevented.

Although CMs highlighted the contribution of emotional wellbeing to CVD, few cited this as a potential avenue for prevention or treatment. Those who did stated that reducing overthinking and avoiding anger can prevent CVD. Respondents more frequently describe the impact of emotional stress when asked how CVD feels—or how someone might know that he or she has CVD—than when interviewers expressly probed for CVD risk factors.

*"When we add salt to food, we should not let it be more, we should let it be less. Those who have the disease, we should tell them not be angry over petty issues. That can prevent it.*

*TZ, others to like rice, millet and beans and all these are healthy foods."*

*"People should stay away from foods that will make you acquire the disease such as oily food and avoid getting excessively hungry and minimize thinking."*

When asked about whether and how they might share their illness with the community, many CMs stated they would eagerly disclose in order to receive help and advice from others. Others, less frequently, mentioned reasons for keeping their illness private, including shyness, fear of becoming a burden to others, privacy concerns or gossip, and stigma.

*"He/she will want others to know his/her condition in order to obtain help from them by way of advice on how to get treatment and also to help others to prevent such conditions in future by sharing with them how he/she acquired the sickness."*

*"It depends. Some people can talk and can't keep secret. All they would do is to spread the news of one's illness hence making it difficult for some patients to publicly talk about their illness."*

## Theme 2: CM impressions about CHOs and CHPS

This theme refers to how CMs interact with the Community Health Officers (CHOs)—front-line nurse providers—at CHPS Compounds regarding CVD. While a few interviewees had not interacted with CHOs enough to comment, many CMs stated CHOs would be capable of diagnosing CVD, using tools like blood pressure cuffs. However, most CMs also believe CHOs are limited by lack of training and medication availability in their capacity to provide comprehensive treatment.

Respondents often stated that in the absence of adequate CVD care at the CHPS compound, the best source of CVD treatment was doctors at the hospital. In practice, community members seek out CVD care due to an acute complaint; CHOs diagnose and provide medication, if available; and then refer them to the local hospital. Of note, every respondent who sought CHPS treatment for CVD—for themselves or a loved one—had done so after experiencing symptoms. There was no mention of being diagnosed with CVD without seeking out symptomatic treatment. Preventive screening did not arise as part of the current care plan.

> *"They [CHPS CHO] even have machine that they to detect whether one is suffering from hypertension or diabetes just that they don't have drugs hence the referral."*

> *"They [CHPS CHO] identify the disease and if cannot treat you they make a referral so if they are not there you can't know you this disease or illness in you. So it is okay for us."*

> *"If you don't tell them [CHPS CHO] that you have heart burns or your heart is beating faster they cannot know what is wrong with you. If you just go and tell them you don't feel well how will they know what is wrong with you.They will ask you, where do you feel the pain? And you will say your heart. And they will ask again, what is wrong with your heart? Then you tell them you are feeling a burning sensation in the heart. So if you the heart is beating up fast is different from if you feel a burning sensation but it is all from the heart so if you don't tell them they won't know."*

> *"With the nurses if you go and it is a heart disease they will give drugs and refer you to the hospital in Paga because they don't have the a lot of drugs here."*

Although most CMs described key behavioral changes (reducing smoking, drinking, salt, and anger) as part of CVD prevention, CMs did not describe such changes as part of the current CHPS CVD prevention plan, although a few described them as part of treatment. However, many interviewees expressed interest in this type of counseling, and most mentioned the need for broader educational efforts in the community.

> *"I would want us to talk about what heart disease is, the symptoms, ways to prevent them, lifestyle modification, diet, exercise, alcohol and the rest and maybe go into the community organize a durbar and you educate them on that."*

> *"They [CHPS CHO] will treat the person by giving drugs for the person to take In order to bring the blood pressure down and tell the person what kinds of food they should eat."*

> *"What do you think that these CHO should be able to do for heart patients that they cannot do at present?" (Interviewer)*

> *"I think these CHO can assist patients with counseling them on what to eat and what not to eat. For instance, counsel them on not be drinking alcohol and smoking but rather they should eat healthy diet to prevent these diseases."*

Interviews with CMs confirmed several key barriers to care described previously, including lack of CVD training, improper medication stockage and inadequate means of transportation [14]. Another challenge was that CHPS nurses did not tend to stay in their assigned catchment areas; two CMs suggested offering housing or better incentives for nurses to stay. Importantly, though, with proper training and resources provided, CMs trusted that CHOs can provide adequate CVD care.

> *"The training should be practical and they should go a bit detailed in to how to manage, provide primary care to people with the heart disease instead of just referring. So that maybe*

*patients who have ischemic attack due to hypertension, how this patient will be stabilized before referring. They should also be taught how to manage severe cases. Personal relations with the patients and then not just their training but they should also be given incentives to stay in the community."*

*". . .Everything without transportation you know, it cannot work so there is the need for them to provide. . . they should establish or they should build new health facilities."*

## Discussion

Our study aimed not only to understand the belief systems of community members in a rural, under-served region of Ghana regarding CVD—but also to situate these views in the context of the existing structure of care and its potential deficits. Previous interviews with communities in sub-Saharan Africa regarding CVD have reported gaps in knowledge of common CVD risk factors, especially relative to Western disease models of conditions such as atherosclerosis [30–32].

Contrary to this work, we found that many respondents readily cited numerous widely-accepted behavioral risk factors for heart disease, including poor diet, tobacco use, and emotional stressors. Yet these same respondents did not describe any model of preventing CVD through medical care, or any experience with asymptomatic screening for CVD. Further, they felt that CVD typically causes sudden death without warning. This seeming contradiction may be due to the existing structure of CVD care in the region, in which patients become aware of disease only in its late stages and care resources are limited. In this schema, prevention is paramount—since the disease is severe, often undiagnosed, and largely untreatable even at diagnosis. But preventative interventions are currently outside the purview of the CHPS nurse or doctor, whose role is to detect and then try to palliate the incurable disease once it manifests symptomatically.

Research into community perspectives on the cardiovascular disease belief models is infrequent in Ghana, and our study is the first to our knowledge to explore community perspectives on the quality of CVD care available at CHPS compounds [33, 34]. Though our study did not aim to quantify or score community members' ability to recognize CVD symptoms—seeking instead to simply describe how CVD manifests in their view–respondents' replies in this domain also contrast with prior research. For example, a 2017 systematic review of knowledge and awareness of CVD risk in sub-Saharan Africa populations across 20 studies reported that up to 73% of respondents could not name a single symptom or risk factor for CVD, with symptoms such as chest pain and weakness in particular infrequently named [31]. On the contrary, nearly all respondents in Navrongo could name at least one CVD symptom such as chest pain—with fatigue and breathing difficulty also commonly noted. With respect to beliefs on access to CVD care, however, our finding of widespread demand for more accessible care—including counseling and medication—aligns with prior data. A 2019 qualitative study of patients' experiences on hypertension care access in rural Bangladesh, Pakistan, and Sri Lanka similarly reported physical distance to care and lack of medication as the main barriers to treatment, and demanded more local access to treatment and counseling [35].

Previous research on community engagement with CHPS [14, 36] suggests that most see its chief role as maternal and child health, and infectious disease care. Our data suggest that CMs nonetheless see CHPS as a viable site for CVD care once specific training is provided and resources are allocated. One such training and resource intervention model to deliver such care is the HEARTS protocol [15], which proposes guidelines for non-physician medical providers (such as CHPS' nurse community health officers, or CHOs, in Navrongo) to treat

hypertension through medication. Yet even nurse-led hypertension care through CHPS may be insufficient, given CHOs have many other care duties and may have limited time to counsel patients [14, 17]. Considerable prior research demonstrates that community health workers (CHWs), typically trained laypersons or volunteers, are impactful in CVD education in many settings [37] and can independently significantly improve hypertension control [38]. CHPS also employs community health volunteers (CHVs) to support CHOs' maternal and child health interventions through home educational visits [39]. Though our current findings did not address the role of these volunteers in CVD care in Navrongo, this cadre of workers is likely also essential to any such future intervention.

Our findings further suggest some details of how such an intervention might operate—especially in concert with previous interviews with CHPS providers. Although community members had a rich understanding of multiple risk factors for cardiac death, ideas regarding how to mitigate existing risk arose but medication was rarely mentioned. Education regarding the potential benefits of screening and treating asymptomatic cardiovascular risk factors such as hypertension may be essential. This work may include both raising awareness of why and how to undergo cardiovascular risk screening as well as available treatments for persons with elevated risk. These treatments, in turn, may include not only medication itself but counseling on how to adhere to a medication regimen—or, analogously, not only advice to quit smoking but practical tools on how to achieve tobacco cessation [40]. In a community in which persons typically seek cardiovascular care only at a symptomatic stage of disease education around how and why medication and lifestyle can disrupt and prevent these conditions is especially essential. Culturally-appropriate strategies for community outreach—such as "durbar" meetings inviting all villagers—may be particularly helpful.

Community members' frequent references to emotional as well as behavioral risk factors—in particular related to depression and anxiety—further reinforce the potential benefit of lifestyle counseling for behavior change. Strategies for such behavior change, frequently designed to treat common mental illnesses such as depression [41] have since been applied to address cardiovascular risk factors such as alcohol or tobacco abuse, or hypertension or obesity [40, 42, 43]. Globally, behavioral risk factors including those aforementioned contribute to 60% of CVD deaths, therefore targeting them is at the crux of any CVD intervention [44]. Moreover, the relationship between mental illness *per se* and cardiovascular disease is well-documented [45], suggesting that intervening on conditions like depression may itself mitigate cardiovascular disease risk in this community.

In contrast to previous discussion with CHPS providers, our findings do not suggest stigma or fear as a significant barrier to seeking CVD care [14]. A majority of respondents stated that they would visit a CHPS compound if they began to experience heart-related symptoms. Yet community members aligned with CHPS providers in highlighting multiple logistical and practical barriers to CHPS cardiovascular disease care [14]. These include limitations intrinsic to clinics, including lack of medication as well as supplies such as blood pressure cuffs, but also challenges in accessing the clinics themselves via available transportation. If education and engagement with communities succeeds in generating further demand for CHPS cardiovascular care and education, supplies and resources available through CHPS infrastructure must be available to meet this demand—at present, fully functional CHPS care is inconsistently accessible across Ghana [36], but ongoing initiatives aim to expand it [46, 47].

Our approach has several limitations. The four analysts coded the interviews separately when creating the initial codebook in order to minimize bias; however, our analyses are subjective due to the qualitative nature of in-depth interviews. The interviews were conducted in the local languages of Kasem and Nankam, which ensured that English would not present a language barrier when gathering data. The interviews, however, were translated to English by

the interviewers for our analyses, which might have introduced some bias or lack of clarity. Further, the coding and thematic analyses of these data were led by researchers not personally familiar with local belief models of CVD in this community [IHM, BP, DJH, AS, KJ, etc.] albeit in close collaboration with researchers with deep personal and professional experience working and living in the Navrongo area [RA, DA, AO, etc.] This study represents the opinions of just 30 community members in the Kassena Nankana East and West districts. We used broad, open-ended questions in order to allow the interviewee to guide the conversation; however, at times, we probed certain topics such as perceived stigma in a way that might have forced certain responses from CMs. Additionally, our interview guide [S1 Appendix] did not define explicitly "heart disease" a priori, potentially creating ambiguity regarding whether the discourse refers to specific heart diseases (such as hypertension); or to all heart diseases; or to deaths from cardiovascular causes.

Despite these limitations, our analysis offers insights on community perception of CVD—how it manifests and how it can be prevented—as well as local understanding of what CHPSs nurse can and cannot provide for them to manage it -in an under-served and under-studied region with a large and rising CVD burden [6]. These interviews can directly inform both the structure and content of an intervention to educate on and mitigate cardiovascular risk factors such as hypertension—by suggesting strategies for preventive education and care; the content of key counseling messages; and the practical barriers to solve for treatment; among other factors.

Our study adds to a limited evidence base for constructing feasible, community-accepted nurse-led interventions for hypertension and other cardiovascular disease in low-resource settings such as Ghana–a leading cause of death and disability both there and worldwide. Although other work has demonstrated limited knowledge of CVD risk factors in similar contexts [31, 32], our work revealed a widespread and robust understanding of such factors, including not only widely known hazards such as smoking, but also more subtle risks such as emotional strain. Yet despite such preventive knowledge, community members in Navrongo often felt CVD disease and death was rapid and hard to control–perhaps due to limited current resources for its prevention and care. Like the CHPS nurses that serve them, community members felt that the local prevention and treatment of CVD is nonetheless feasible with future action, and detailed specific means for achieving it. This work has directly informed a pilot intervention for the screening and control of hypertension and depression in Navrongo through CHPS, whose implementation and evaluation is now ongoing.

## Supporting information

**S1 Appendix. Interview guide, community member.**
(PDF)

**S1 File. Manuscript inclusivity questionnaire.**
(PDF)

## Acknowledgments

The authors gratefully thank the Navrongo Health Research Centre and all of its staff and colleagues for hosting this work, and our colleagues and staff from Columbia University, New York University, the Arnhold Institute for Global Health, and the Icahn School of Medicine at Mount Sinai. We separately thank the communities in and around Navrongo, Ghana who participated in this research with our study team.

## Author Contributions

**Conceptualization:** Bhavana Patil, Isla Hutchinson Maddox, Raymond Aborigo, James F. Phillips, David J. Heller.

**Data curation:** Bhavana Patil, Isla Hutchinson Maddox, Denis Awuni.

**Formal analysis:** Bhavana Patil, Isla Hutchinson Maddox, Raymond Aborigo, Allison P. Squires, Carol R. Horowitz, Khadija R. Jones, David J. Heller.

**Funding acquisition:** Abraham R. Oduro, James F. Phillips, David J. Heller.

**Investigation:** Bhavana Patil, Isla Hutchinson Maddox, Raymond Aborigo, Carol R. Horowitz, David J. Heller.

**Methodology:** Bhavana Patil, Isla Hutchinson Maddox, Raymond Aborigo, Allison P. Squires, Carol R. Horowitz, James F. Phillips, David J. Heller.

**Project administration:** Denis Awuni, Abraham R. Oduro, James F. Phillips, Khadija R. Jones, David J. Heller.

**Resources:** Denis Awuni, Abraham R. Oduro, James F. Phillips, Khadija R. Jones, David J. Heller.

**Software:** Khadija R. Jones, David J. Heller.

**Supervision:** Raymond Aborigo, Allison P. Squires, Denis Awuni, James F. Phillips, Khadija R. Jones, David J. Heller.

**Validation:** Raymond Aborigo, Allison P. Squires.

**Writing – original draft:** Bhavana Patil, Isla Hutchinson Maddox.

**Writing – review & editing:** Bhavana Patil, Isla Hutchinson Maddox, Raymond Aborigo, Allison P. Squires, Denis Awuni, Carol R. Horowitz, James F. Phillips, Khadija R. Jones, David J. Heller.

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
