## [Decision Letter · Decision Letter 0]

6 Oct 2022

PONE-D-21-40120Community Perspectives on Cardiovascular Disease Control in Rural Ghana: A Qualitative StudyPLOS ONE

Dear Dr. Heller,

Thank you for submitting your manuscript to PLOS ONE. After careful consideration, we feel that it has merit but does not fully meet PLOS ONE’s publication criteria as it currently stands. Therefore, we invite you to submit a revised version of the manuscript that addresses the points raised during the review process.

In general. the manuscript is well written and presents valuable information about community perspectives of cardiovascular diseases (CVD) care in Ghana. In addition to responding to reviewer comments below, please revisit the use of the term “subjects” to refer to study participants. Please refer yourself to the following publication about respectful terms for study volunteers. Chalmers I. People are "participants" in research. Further suggestions for other terms to describe "participants" are needed. BMJ. 1999 Apr 24;318(7191):1141. doi: 10.1136/bmj.318.7191.1141a. PMID: 10213744; PMCID: PMC1115535.

We look forward to receiving your revised manuscript.

Kind regards,

Dorina Onoya

Academic Editor

PLOS ONE

Journal Requirements:

3. Please include a complete copy of PLOS’ questionnaire on inclusivity in global research in your revised manuscript. Our policy for research in this area aims to improve transparency in the reporting of research performed outside of researchers’ own country or community. The policy applies to researchers who have travelled to a different country to conduct research, research with Indigenous populations or their lands, and research on cultural artefacts. The questionnaire can also be requested at the journal’s discretion for any other submissions, even if these conditions are not met.  Please find more information on the policy and a link to download a blank copy of the questionnaire here: https://journals.plos.org/plosone/s/best-practices-in-research-reporting. Please upload a completed version of your questionnaire as Supporting Information when you resubmit your manuscript.

“This work was supported by the Fogarty International Center of the National Institutes of Health, USA (R21 TW010452-01 to JFP, ARO, and DJH); the Arnhold Institute for Global Health at the Icahn School of Medicine at Mount Sinai, USA; the Navrongo Health Research Centre, Ghana; and Teva Pharmaceutical Industries, Israel (DJH).

5. Thank you for providing the following Funding Statement: 

“DJH reports: I have read the journal's policy and the authors of this manuscript have the following competing interests: research funding from Teva Pharmaceutical Industries.

This funder had no role in study design, data collection and analysis, decision to publish, or preparation of the manuscript.”

We note that one or more of the authors is affiliated with the funding organization, indicating the funder may have had some role in the design, data collection, analysis or preparation of your manuscript for publication; in other words, the funder played an indirect role through the participation of the co-authors.

If the funding organization did not play a role in the study design, data collection and analysis, decision to publish, or preparation of the manuscript and only provided financial support in the form of authors' salaries and/or research materials, please review your statements relating to the author contributions, and ensure you have specifically and accurately indicated the role(s) that these authors had in your study in the Author Contributions section of the online submission form. Please make any necessary amendments directly within this section of the online submission form.  Please also update your Funding Statement to include the following statement: “The funder provided support in the form of salaries for authors [insert relevant initials], but did not have any additional role in the study design, data collection and analysis, decision to publish, or preparation of the manuscript. The specific roles of these authors are articulated in the ‘author contributions’ section.”

If the funding organization did have an additional role, please state and explain that role within your Funding Statement.

Please also provide an updated Competing Interests Statement declaring this commercial affiliation along with any other relevant declarations relating to employment, consultancy, patents, products in development, or marketed products, etc. 

Reviewers' comments:

Reviewer's Responses to Questions

**Comments to the Author**

1. Is the manuscript technically sound, and do the data support the conclusions?

Reviewer #1: Partly

Reviewer #2: Yes

2. Has the statistical analysis been performed appropriately and rigorously? 

Reviewer #1: I Don't Know

Reviewer #2: Yes

3. Have the authors made all data underlying the findings in their manuscript fully available?

Reviewer #1: No

Reviewer #2: Yes

4. Is the manuscript presented in an intelligible fashion and written in standard English?

Reviewer #1: Yes

Reviewer #2: Yes

5. Review Comments to the Author

Reviewer #1: Heller et al report a qualitative study from Ghana on community members’ awareness of CVD and their views on seeking care from community health nurses.

CMs described CVD as an acute illness and had poor awareness regarding chronic CVD. They expressed the desire for CVD behavioral education from community nurses at local clinics, but feel the local health system is now too fragile to offer other CVD interventions.

The study is on an important topic as CVD is rising at a rapid pace in SSA. Community health worker-led models of care have been shown to be effective in LMIC settings on BP lowering and CVD risk reduction. Evidence from Ghana is needed, and stakeholder assessment if the first step even before a feasibility study to adapt evidence-based strategies to the local context.

However, there are several concerns about the introduction, methods, analysis and interpretation that need to be addressed.

1. It is advisable to use a framework for the qualitative work that is aligned with the objectives of the study. However, this was not done which is acknowledged by the authors. What steps were taken during the analysis to ensure that the themes that emerged were captured adequately?

2. If the data collectors were not trained in CVD model. It is vital that the interviewers are trained in probing as indicated during suggestive answers by the respondent. How were the data collectors trained?

3. The probes were in English and not translated into the local languages. How was appropriate translation of technical words ensured

4. How was data saturation ensured?

5. Did the community members suffer from CVD? Was any stratification used during sampling? What was the gender balance?

6. Please discuss how do your findings compare with those from other in SSA and in other LMICs

https://journals.plos.org/plosone/article?id=10.1371/journal.pone.0189264

https://journals.plos.org/plosone/article?id=10.1371/journal.pone.0211100

7. In the introduction and discussion the authors underscore the importance of a “community nurse” led intervention in LMIC setting. However, much of the evidence is about community health worker (CHW)-led interventions. Please make a distinction. There is a shortage of nurses in LMICs. Task-shifting/sharing models of hypertension care are unlikely to be scalable unless led by CHW as shown in multiple studies. Please cite these studies: https://www.nejm.org/doi/10.1056/NEJMoa1911965?url_ver=Z39.88-2003&rfr_id=ori:rid:crossref.org&rfr_dat=cr_pub%20%200pubmed

Reviewer #2: The manuscript by Patil et al. entitled 'Community Perspectives on Cardiovascular Disease Control in Rural Ghana: A

Qualitative Study' is well written and provides support for integration of CVD services into CHPS.

I have a few minor comments.

1. The following statement is unclear on line 156 "“We did not use any a priori framework to parse or categorize the findings data. However, our approach, like the interview content itself, builds on prior work’s. For instance, insofar as that work demonstrates that CHPS nurses believe community members CVD lack knowledge of the causes and treatments for we expressly probed these throughout the semi-structured interview.” I am not a qualitative researchers. Is the fact that it builds on others work not a priori framework in some way? I assume that the previous work you cite helped you identify some themes in your own work in advance?

2. Line 171, can you put in a sentence about how one would interpret a kappa statistic of 0.68?

3. line 176 can you add in gender break down as well?

6. PLOS authors have the option to publish the peer review history of their article (what does this mean?). If published, this will include your full peer review and any attached files.

Reviewer #1: No

Reviewer #2: No

---

## [Author Response · Author response to Decision Letter 0]

18 Nov 2022

18 November 2022

Dear Dr. Onoya and colleagues, 

My co-authors and I thank you for your review of our manuscript, “Community Perspectives on Cardiovascular Disease Control in Rural Ghana: A Qualitative Study.” We have responded to each of the reviewers’ requested revisions and clarifications. 

See attached, separately, a copy of the manuscript with all changes tracked; and a separate clean copy with these changes incorporated. Our point-by-point response to your comments and the reviewers’ is detailed below, and also in a separate Microsoft Word document labeled "Response to Reviewers."

We look forward to your further review.

Best regards, 

David Heller MD MPH

david.heller@mssm.edu

Editorial Comments

In general. the manuscript is well written and presents valuable information about community perspectives of cardiovascular diseases (CVD) care in Ghana. In addition to responding to reviewer comments below, please revisit the use of the term “subjects” to refer to study participants. Please refer yourself to the following publication about respectful terms for study volunteers. Chalmers I. People are "participants" in research. Further suggestions for other terms to describe "participants" are needed. BMJ. 1999 Apr 24;318(7191):1141. doi: 10.1136/bmj.318.7191.1141a. PMID: 10213744; PMCID: PMC1115535.

We appreciate this comment and agree that the term “subjects,” is problematically demeaning, as argued above. We have replaced the two instances where we used “subjects” with “participants” – see the revised “Sample and Recruitment” section. We use the word “participant” in all other revisions to the manuscript as well. 

Provide additional details regarding participant consent. In the ethics statement in the Methods and online submission information, please ensure that you have specified (1) whether consent was informed and (2) what type you obtained (for instance, written or verbal, and if verbal, how it was documented and witnessed). If your study included minors, state whether you obtained consent from parents or guardians. If the need for consent was waived by the ethics committee, please include this information.

We thank you for underscoring this point. We have revised the “Data Collection” section to expressly clarify how we obtained informed verbal consent and documented this result. We also affirm that no participants were minors. We have also revised our online submission to confirm these facts. 

3. Please include a complete copy of PLOS’ questionnaire on inclusivity in global research in your revised manuscript. Our policy for research in this area aims to improve transparency in the reporting of research performed outside of researchers’ own country or community. The policy applies to researchers who have travelled to a different country to conduct research, research with Indigenous populations or their lands, and research on cultural artefacts. The questionnaire can also be requested at the journal’s discretion for any other submissions, even if these conditions are not met. Please find more information on the policy and a link to download a blank copy of the questionnaire here:https://journals.plos.org/plosone/s/best-practices-in-research-reporting. Please upload a completed version of your questionnaire as Supporting Information when you resubmit your manuscript.

We have completed this questionnaire and attached it to our online resubmission as Supporting Information. 

Thank you for stating in your Funding Statement: “This work was supported by the Fogarty International Center of the National Institutes of Health, USA (R21 TW010452-01 to JFP, ARO, and DJH); the Arnhold Institute for Global Health at the Icahn School of Medicine at Mount Sinai, USA; the Navrongo Health Research Centre, Ghana; and Teva Pharmaceutical Industries, Israel (DJH). The funders had no role in study design, data collection and analysis, decision to publish, or preparation of the manuscript.” Please provide an amended statement that declares *all* the funding or sources of support (whether external or internal to your organization) received during this study, as detailed online in our guide for authors at http://journals.plos.org/plosone/s/submit-now. Please also include the statement “There was no additional external funding received for this study.” in your updated Funding Statement. Please include your amended Funding Statement within your cover letter. We will change the online submission form on your behalf.

“DJH reports: I have read the journal's policy and the authors of this manuscript have the following competing interests: research funding from Teva Pharmaceutical Industries. This funder had no role in study design, data collection and analysis, decision to publish, or preparation of the manuscript.” We note that one or more of the authors is affiliated with the funding organization, indicating the funder may have had some role in the design, data collection, analysis or preparation of your manuscript for publication; in other words, the funder played an indirect role through the participation of the co-authors.

If the funding organization did not play a role in the study design, data collection and analysis, decision to publish, or preparation of the manuscript and only provided financial support in the form of authors' salaries and/or research materials, please review your statements relating to the author contributions, and ensure you have specifically and accurately indicated the role(s) that these authors had in your study in the Author Contributions section of the online submission form. Please make any necessary amendments directly within this section of the online submission form. Please also update your Funding Statement to include the following statement: “The funder provided support in the form of salaries for authors [insert relevant initials], but did not have any additional role in the study design, data collection and analysis, decision to publish, or preparation of the manuscript. The specific roles of these authors are articulated in the ‘author contributions’ section. If the funding organization did have an additional role, please state and explain that role within your Funding Statement.

We thank the editor for requesting these clarifications. We have revised the Funding Statement within the manuscript, as well as the Manuscript Data section of the online submission form, to detail all the matters above – for example affirming that we have named all funders and separately that funders provided salary and material support but had no role in any aspects of the creation or writing of the manuscript itself. 

Please also provide an updated Competing Interests Statement declaring this commercial affiliation along with any other relevant declarations relating to employment, consultancy, patents, products in development, or marketed products, etc. 

We have separately revised the Competing Interests Statement to clarify the exact role of our one commercial for-profit funder (Teva Pharmaceutical Industries), and to expressly affirm we have no other conflicts to declare in any of the other domains detailed above.

Within your Competing Interests Statement, please confirm that this commercial affiliation does not alter your adherence to all PLOS ONE policies on sharing data and materials by including the following statement: "This does not alter our adherence to PLOS ONE policies on sharing data and materials.” (as detailed online in our guide for authorshttp://journals.plos.org/plosone/s/competing-interests). If this adherence statement is not accurate and there are restrictions on sharing of data and/or materials, please state these. Please note that we cannot proceed with consideration of your article until this information has been declared.

We have adjusted our Competing Interest Statement to separately affirm that our affiliation with Teva Pharmaceutical Industries does not alter or otherwise infringe on any aspect of adherence to PLOS ONE policies regarding sharing data and materials, using the language above. 

Please include captions for your Supporting Information files at the end of your manuscript, and update any in-text citations to match accordingly. Please see our Supporting Information guidelines for more information:http://journals.plos.org/plosone/s/supporting-information.

We have amended the end of the manuscript to include a caption for our supplemental item (the interview guide), and referenced this caption in the text twice as well. In each case, the file is named “S1 Appendix” per the formatting request linked above. 

Reviewer 1

It is advisable to use a framework for the qualitative work that is aligned with the objectives of the study. However, this was not done which is acknowledged by the authors. What steps were taken during the analysis to ensure that the themes that emerged were captured adequately?

We thank the reviewer for requesting this important clarification. We have addended the Design and Data Analysis elements of the Methods section extensively to clarify and justify the precise approach we took. See details in the manuscript – but in brief, we selected a qualitative descriptive approach to our interview guide and analytic approach precisely because it avoids a pre-existing framework of interpretation that could have biased how we collected our data and how we generated codes and themes during analysis. However, we used conventional content analysis – a validated approach to interpreting data from qualitative descriptive studies – to ensure we systematically and exhaustively agreed on how to generate a definitive codebook to turn the transcribed data into codes. Once we thereby coded all data, following calculation of a Kappa statistic for inter-reviewer agreement, four authors independently reviewed all these codes to generate cross-cutting themes, which we discussed and agreed on iteratively before reaching final agreement. Of note, although our approach avoided an a priori interpretive framework, we did use an agreed model to analyze data once coded, namely the socio-ecological model – and we did use findings from our prior published work to shape our analysis. These details, with references, are now explained at length in the Data Analysis element of the Methods section.

If the data collectors were not trained in CVD model. It is vital that the interviewers are trained in probing as indicated during suggestive answers by the respondent. How were the data collectors trained?

We thank the reviewer for requesting this clarification as well. Please see the revised Data Collection element of the Methods section, where we detail that two co-authors from the Navrongo Health Research Centre trained all interviewers on matters such as how to probe for further information in an open and unbiased manner; how to convey technical or medical terms; and how to obtain informed consent among other matters. 

The probes were in English and not translated into the local languages. How was appropriate translation of technical words ensured. 

As noted above, we have amended the Data Collection section of the manuscript to address this issue. Firstly, we actively avoided any technical terms in the interview guide (see Appendix) to prevent inherent confusion or language or translation errors. However, we did translate the interview guide orally into the local languages Kasem and Nankam when conducting the interview in these languages as per the preference of the participant. As detailed above, all interviewers were fluent in at least one of these two languages. Audio recordings of the interviews were then translated back into English by the same trained interviewers. We acknowledge in the Limitations section that, despite their language fluency, interviewers may have introduced subconscious bias or translation error during this process. 

How was data saturation ensured?

We have amended the section titled “Sample & Recruitment” to clarify this matter. Specifically, we targeted approximately 30 participants based on prior empiric research (see references 20 and 21) demonstrating that between 12 and 31 interviews is sufficient to achieve data saturation. We also detail here precisely how we defined data saturation, per Saunders et al (reference 21). We have separately amended the first paragraph of the Results section to affirm that data saturation was, in fact, achieved.

Did the community members suffer from CVD? Was any stratification used during sampling? What was the gender balance?

We appreciate the comment above and have addressed it in the Sample & Recruitment section as follows:

“Our aim was to understand the perceptions of the community regarding CVD (e.g., family members of persons living with cardiovascular disease) and not just those diagnosed with CVD. Therefore, we did not actively screen participants’ medical history.”

In short, we did not stratify for CVD diagnosis, nor actively seek to enrich our participant sample with persons living with CVD for the reasons above. However, we separately clarify in this section that CVD is nonetheless very common in the Navrongo communities where our research occurred. See references 6 and 24, which demonstrate respectively that CVD causes some 14% of all mortality in the community, and that 25% of middle-aged adults meet criteria for hypertension.

Please discuss how do your findings compare with those from other in SSA and in other LMICs.

We thank the reviewer for noting these references, which highlight how our findings compare with the results of this prior work – in particular, a systemic review of hypertension knowledge in sub-Saharan Africa, and a qualitative study of beliefs regarding hypertension control in Southeast Asia. Of note, our findings contrast with the first of these studies, but not the second. Please see the third paragraph of the Discussion section, as follows: 

“Though our study did not aim to quantify or score community members’ knowledge of CVD - seeking instead to simply describe how CVD manifests in their view - our finding of numerous diverse reported symptoms of CVD contrasts with prior research. For example, a 2017 systematic review of knowledge and awareness of CVD risk in sub-Saharan Africa populations across 20 studies reported that up to 73% of respondents could not name a single risk factor for CVD, with symptoms such as chest pain and weakness, and risk factors such as stress and smoking infrequently named [35]. On the contrary, nearly all respondents in Navrongo could name at least one CVD symptom such as chest pain - and stress and tobacco were frequently-cited risk factors. With respect to beliefs on access to CVD care, however, our finding of widespread demand for more accessible care - including counseling and medication - aligns with prior data. A 2019 qualitative study of patients’ experiences on hypertension care access in rural Bangladesh, Pakistan, and Sri Lanka similarly reported physical distance to care and lack of medication as the main barriers to treatment, and demanded more local access to treatment and counseling [36].”

In the introduction and discussion the authors underscore the importance of a “community nurse” led intervention in LMIC setting. However, much of the evidence is about community health worker (CHW)-led interventions. Please make a distinction. There is a shortage of nurses in LMICs. Task-shifting/sharing models of hypertension care are unlikely to be scalable unless led by CHW as shown in multiple studies. Please cite these studies:https://www.nejm.org/doi/10.1056/NEJMoa1911965?url_ver=Z39.88-2003&rfr_id=ori:rid:crossref.org&rfr_dat=cr_pub%20%200pubmed

We agree that the distinction between nurses (as formal health providers) and community health workers is essential in discussion of task-sharing models for cardiovascular disease as with any other form of primary care in LMICs. We also agree that prior literature supports CHWs as effective for and essential to hypertension control (as demonstrated by Jafar et al), and acknowledge that they are therefore likely critical for future such intervention in Navrongo. We have revised the discussion to acknowledge this point in paragraph four: 

“Yet even nurse-led hypertension care through CHPS may be insufficient, given CHOs have many other care duties and may have limited time to counsel patients [14,17]. Considerable prior research demonstrates that community health workers (CHWs), typically trained laypersons or volunteers, are impactful in CVD education in many settings [37] and can independently significantly improve hypertension control [38]. CHPS also employs community health volunteers (CHVs) to support CHOs’ maternal and child health interventions through home educational visits [39]. Though our current findings did not address the role of these volunteers in CVD care in Navrongo, this cadre of workers is likely also essential to any such future intervention.”

Reviewer 2

The following statement is unclear on line 156 "“We did not use any a priori framework to parse or categorize the findings data. However, our approach, like the interview content itself, builds on prior work’s. For instance, insofar as that work demonstrates that CHPS nurses believe community members CVD lack knowledge of the causes and treatments for we expressly probed these throughout the semi-structured interview.” Is the fact that it builds on others work not a priori framework in some way? I assume that the previous work you cite helped you identify some themes in your own work in advance?

We thank the reviewer for highlighting this important distinction, which Reviewer 1 has also raised in the comments above regarding how we achieved analytic rigor. We have amended the Design and Data Analysis elements of the Methods section of the manuscript to clarify the issue. In short, we now clarify in the Design section the deliberate reason why we did not use any a priori framework in creating themes (namely, to avoid introduction of bias from the presumptions inherent to any such framing). However, we separately note in the Data Analysis section that our analysis – while relying only on the data in its own right to inductively generate themes (i.e., conventional content analysis), rather than prior thematic frameworks– was informed by the socio-ecological model as well as by our prior research. These concepts did not *create* our themes, but did indeed shape the way in which we interpreted the coded data to generate those themes. We hope that this language clarifies this subtle but important distinction.

Line 171, can you put in a sentence about how one would interpret a kappa statistic of 0.68?

We have provided a better framework to introduce a Kappa statistic including how ratings are scaled according to traditional guidelines. Please see the second paragraph of the Data Analysis element of the Methods section, where we clarify as per below that we achieved “substantial agreement” between our two main coders IHM and BP: 

“To ensure inter-rater reliability between these reviewers, we calculated a Kappa score of agreement between the two sets of independent codes they applied to select transcripts chosen at random, with a result of 0.68. Kappa scores by definition range from -1 to +1, with a higher score indicating greater agreement - specifically, a score of 0.68 indicates “substantial agreement.”

line 176 can you add in gender break down as well? 

We have edited this section to indicate that we actively recruited approximately equal numbers of persons identifying as male or female, but did not record the final result.

---

## [Decision Letter · Decision Letter 1]

28 Dec 2022

Community perspectives on cardiovascular disease control in rural Ghana: A qualitative study

PONE-D-21-40120R1

Dear Dr. David J Heller,

We’re pleased to inform you that your manuscript has been judged scientifically suitable for publication and will be formally accepted for publication once it meets all outstanding technical requirements.

Kind regards,

Dorina Onoya

Academic Editor

PLOS ONE

Reviewers' comments:

Reviewer's Responses to Questions

**Comments to the Author**

1. If the authors have adequately addressed your comments raised in a previous round of review and you feel that this manuscript is now acceptable for publication, you may indicate that here to bypass the “Comments to the Author” section, enter your conflict of interest statement in the “Confidential to Editor” section, and submit your "Accept" recommendation.

Reviewer #1: All comments have been addressed

Reviewer #2: (No Response)

2. Is the manuscript technically sound, and do the data support the conclusions?

Reviewer #1: Yes

Reviewer #2: Yes

3. Has the statistical analysis been performed appropriately and rigorously? 

Reviewer #1: Yes

Reviewer #2: Yes

4. Have the authors made all data underlying the findings in their manuscript fully available?

Reviewer #1: Yes

Reviewer #2: Yes

5. Is the manuscript presented in an intelligible fashion and written in standard English?

Reviewer #1: Yes

Reviewer #2: Yes

6. Review Comments to the Author

Reviewer #1: The revisions are satisfactory. No further queries. The current version is acceptable for publication.

Reviewer #2: (No Response)

7. PLOS authors have the option to publish the peer review history of their article (what does this mean?). If published, this will include your full peer review and any attached files.

Reviewer #1: No

Reviewer #2: No

---

## [Editor Report · Acceptance letter]

10 Jan 2023

PONE-D-21-40120R1 

Community perspectives on cardiovascular disease control in rural Ghana: A qualitative study 

Dear Dr. Heller:

I'm pleased to inform you that your manuscript has been deemed suitable for publication in PLOS ONE. Congratulations! Your manuscript is now with our production department. 

Kind regards, 

on behalf of

Dr. Dorina Onoya 

Academic Editor

PLOS ONE